# Phage Display Technology in Biomarker Identification with Emphasis on Non-Cancerous Diseases

**DOI:** 10.3390/molecules29133002

**Published:** 2024-06-25

**Authors:** Mohammad Sadraeian, Reza Maleki, Mahta Moraghebi, Abasalt Bahrami

**Affiliations:** 1Institute for Biomedical Materials and Devices (IBMD), Faculty of Science, University of Technology Sydney, Sydney, NSW 2007, Australia; mohammad.sadraeian@uts.edu.au; 2Adelaide Medical School, University of Adelaide, Adelaide, SA 5005, Australia; 3Department of Chemistry and Biochemistry, Bioengineering, and Materials Science and Engineering, University of California, Los Angeles, CA 90095, USA

**Keywords:** phage display, biopanning, diagnosis, therapeutic applications

## Abstract

In recent years, phage display technology has become vital in clinical research. It helps create antibodies that can specifically bind to complex antigens, which is crucial for identifying biomarkers and improving diagnostics and treatments. However, existing reviews often overlook its importance in areas outside cancer research. This review aims to fill that gap by explaining the basics of phage display and its applications in detecting and treating various non-cancerous diseases. We focus especially on its role in degenerative diseases, inflammatory and autoimmune diseases, and chronic non-communicable diseases, showing how it is changing the way we diagnose and treat illnesses. By highlighting important discoveries and future possibilities, we hope to emphasize the significance of phage display in modern healthcare.

## 1. Introduction

The concept of phage display was originally proposed by George P. Smith who used filamentous bacteriophages M13 to insert protein-coding genes and called the process of adopting a specific protein “panning” [1]. Phage display is a cost-efficient technique used to identify peptides that serve to enhance antigen–antibody interactions. The integration of gene engineering and combinatorial chemistry has been carried out in the phage display technology that utilizes exogenous DNA to be interested in phage coat encoding sequences as well as filamentous phage genomes [2]. After being expressed, phage particles are generated which carry the coat protein of the phage combined with the sequence of interest. In this manner, these sequences are able to interact with a variety of external targets or ligands [3]. There are three general categories of phage display vectors. Type 3 and 33 are two of the most common phage display vectors that have been developed. There is one copy of the PIII gene in the type 3 vector, and the peptide fragment is cloned in a frame with the PIII gene. The type 33 vector contains two copies of the PIII gene, one wild type and one synthetic. Unlike the synthetic PIII gene, which employs a periplasmic signal sequence in wild type, the signal sequence consists of an endogenous 18 amino acids. In addition, the wild-type PIII is not fused to any companion, in contrast to the synthetic PIII, which is employed as a fusion partner [4]. In addition to the type 3 and 33 vectors, a phagemid vector (so-called type 3 + 3) with no size disadvantage has also been developed. As this system is devoid of any additional phage genes, bacteria carrying it must be infected with a helper phage to enable the production of phage fragments [5].

As one of its most noteworthy characteristics, the capacity of phage display to generate a large number of libraries implies that these libraries could be employed in phage display to identify functional proteins [6]. Employing immunization techniques presents challenges in identifying peptides and antigens across various epitope and peptide spectra. However, the implementation of phage display technology offers a solution to these obstacles [7]. On the basis of its versatile structure and profusion of coat proteins (pIII, pVI, pVII, pVIII, and pIX), M13 emerges as the bacteriophage of choice within this context. The pIII coat protein plays a crucial role in enabling the bacteriophage to bind to the bacterial pilus, facilitating the attachment of the virus. Following the binding of pIII to a protein on the bacterial membrane, the phage genome is transported into the bacterium to generate DNA. Subsequently, the required proteins for the production of phage particles will be generated. Although M13 phage particles are released from the bacterium, they do not cause bacterial death. Instead, the bacterium continues to exist in a perpetual state of infection and proliferation [8]. Figure 1 summarizes the process of selecting proteins or peptides by phage display.

Phage display finds applications in various fields including enzyme inhibition, receptor modulation, epitope mapping, and vaccine development [9]. Its versatility and potential for innovation suggest a promising future for the technique. Several past reviews have explored the techniques and applications of phage display libraries, yet none have specifically examined their role in non-cancerous diseases [10,11,12]. Here, we will briefly cover the basics of the technology, improvements made to enhance its effectiveness, and its common applications in non-cancerous diseases, including degenerative diseases, inflammatory and autoimmune conditions, and chronic non-communicable diseases. We also highlight selected therapeutic uses of recombinant antibodies and outline recent developments in phage display technology.

## 2. The Fundamental Steps of Phage Display Technology

### 2.1. Natural and Synthetic Peptide Library

In terms of peptides, phage libraries are either composed of natural peptides or synthetic ones. Natural peptide libraries are constructed by extracting DNA fragments from the organism (like mice) and inserting them into the phage particle and upon inducing expression, natural peptides will express on the surface of phages. On the other hand, synthetic libraries consist of cloned synthetic and arbitrarily created oligonucleotide sequences that are incorporated into the phage genome [13]. In synthetic libraries, complementarity-determining regions (CDRs) are artificially diverse. The CDR sequence diversity could be produced using a random combination of single nucleotides or three nucleotides [14]. Additionally, synthetic and natural libraries can be coupled to create semi-synthetic libraries [15].

### 2.2. Antibodies-Based Libraries

Several small antibody fragments such as scFv (variable light and heavy chain fragments joined by polypeptide linker) and Fab (antigen binding region) have been used to prepare phage libraries. These antibody-based libraries are widely used for epitope mapping, by biopanning of the antigen or the whole cell [16]. The production of these libraries can be achieved through immunization, which entails the collection of antibodies from animals that have received antigens. On the other hand, non-immunized (naïve) libraries are created by integrating antibody fragments obtained from healthy B cells through a specific polymerase chain reaction called splice by overlap extension PCR (SOE-PCR) [17].

### 2.3. Biopanning

Multiple steps are involved in biopanning, including purification, immobilization, washing, phage elution, infection, propagation, and re-selection. Once the bacteriophage viruses have been propagated and used to infect the bacteria, the resultant phages are employed in the biopanning. This method entails exposing the libraries to the antigen of interest that is attached to a solid surface. After incubation, phages that are not bound are washed and discarded, while phages that are bound are harvested and assayed using ELISA (enzyme-linked immunosorbent assay). A new and more specific phage library is created by infecting bacteria with the collected phages from the previous step. This library will be used in the subsequent cycle of biopanning. Using the phages generated in previous rounds, three to five rounds of panning are conducted to obtain phages with high affinity and specificity [18].

### 2.4. The Diagnostic Effectiveness of Phage Display Technology

Numerous recombinant antibodies and peptides against pathogens and cancer antigens have been developed for diagnostic purposes using phage display. Peptides generated by phage display have the potential to be exploited for molecular imaging diagnostics in the field of cancer [19]. These peptides can be employed in imaging various malignancies by conjugating or labeling them with nanoparticles, radionuclides, and fluorescents. The conventional method for tumor imaging is performed by monoclonal antibodies (mAbs) with whole size. Peptides could be more effective than antibodies because of their low molecular weight which provides better tumor penetration and low immunogenicity because of lacking Fc regions [20,21].

In one study, for example, a peptide with specific binding capabilities to the extracellular domain of HER3 (HER3P1) was produced by the phage display technique [22]. Traditional HER3 imaging, which uses a radiolabeled full-size antibody-like patritumab, does not lead to accurate PET imaging since these antibodies exhibit weak tumor penetration and non-specific accumulation in tissues that are not the target [23,24]. It demonstrated that the radiolabeled HER3P1 had a potential for application in clinical imaging as it showed a great specificity and low background uptake by in vivo PET imaging [22].

Employing phage display, a diverse set of recombinant antibodies for pathogen detection (e.g., bacteria and viruses) has been developed. Since this technology comprises an in vitro screening procedure, it has eliminated the limitations and constraints of hybridoma technology [25]. The majority of these antibodies, which are antibody fragments such as scFv, Fab, and nanobodies, are applied in ELISA, immunoblot, immunofluorescent assays, and immunoprecipitation [26].

Although generating monoclonal antibodies is both an expensive and time-consuming task, they are widely deployed in immunological tests. In contrast, phage display allows for the quick collection of recombinant antibodies at a low cost and with the same affinity. Furthermore, in certain investigations, antibodies generated through phage display were more sensitive toward an antigen compared to other traditional techniques. A peptide that binds the spike protein of the transmissible gastroenteritis virus (TGEV) was developed by Suo and colleagues using phage display for diagnostic purposes. It was discovered that phage-mediated ELISA was more sensitive than conventional antibody-based ELISA [27]. 

In a recent investigation, Sulong et al. used phage display to design a VH antibody against capsular polysaccharide (CPS) of *Streptococcus suis* serotype 2 (Figure 2A). It has been demonstrated that this VH antibody has the potential to be a useful diagnostic instrument for *S. suis* serotype differentiation since similar epitopes are shared by serotypes 1, 1/2, 2, and 14 (Figure 2B) [28,29]. The positive phage clone 47B3 displaying VH antibody disclosed the highest binding ability with no cross-reactivity with *S. suis* serotypes 1/2, 1, or 14, whereas prior mAbs against *S. suis* serotype 2 have revealed cross-reactivity with serotypes 2, 1/2, 1, and 14. Furthermore, the pure VH antibody (namely 47B3 VH) has the remarkable ability to discriminate between serotypes 2 and 1/2 effectively (Figure 2C), which is not possible to be detected by PCR-based serotyping [30]. The purpose of this review is to highlight the applications of phage display as a therapeutic and diagnostic tool in non-cancerous diseases. Additionally, a very recent study developed an innovative method to characterize exosomes using phage display technology. This breakthrough system for detecting and isolating disease-related exosomes could be useful in defining novel biomarkers to treat diseases where exosomes are involved [31].

## 3. Applications of Phage Display in Non-Cancer Diseases

Phage display has expanded its applications beyond cancer research, notably in the examination and treatment of non-cancer diseases [6,32]. One important field of study involves infectious diseases, where a phage display has been employed to find and create new peptides or antibodies that target particular pathogens. This method involves presenting collections of peptides or antibodies derived from phages on the surface of these viruses, allowing for the selective screening of molecules that can bind to infectious agents like bacteria or viruses. This method has helped in finding potential markers for diagnosis, developing treatments, and creating vaccine candidates for infectious diseases, thus aiding progress in infectious disease research. Phage display has also demonstrated potential in autoimmune and inflammatory conditions, besides its applications in infectious diseases.

Phage libraries can be employed to identify peptides or antibodies that interact with specific components of the immune system, offering insights into disease mechanisms and potential therapeutic targets. By understanding the molecular interactions involved in autoimmune and inflammatory conditions, we can develop targeted interventions to modulate the immune response and potentially treat diseases such as rheumatoid arthritis [33,34,35,36,37], lupus [38,39,40], and inflammatory diseases [41,42].

### 3.1. Degenerative Diseases

#### Alzheimer’s Disease

Alzheimer’s disease (AD) is classified as a neurological illness that is marked by dementia, which substantially impairs functioning and creates social hardship [43]. The pathogenic processes of AD include the formation of mesh-like plaques and intracellular neurofibrillary tangles by the accumulation of tau protein and amyloid-beta (Aβ) peptides. The majority of the presently accessible medications are designed to address the advanced phase of the illness and have a limited impact on achieving a complete recovery from the disease [44]. Phage display has shown to be a trusted and established platform for drug development for the treatment of AD [45]. Prior research used mirror phage display to isolate the D3 enantiomer from a 9-mer peptide library, which demonstrated the ability to decrease plaque development in the brains of rats [46].

Dammers and colleagues identified a D-enantiomer peptide targeting PHF6 (paired helical fragment 6) within the tau protein, thereby influencing its accumulation process. Their findings suggest that, given its cell-penetrating properties, stability, and low immunogenicity, this peptide merits further investigation for its potential applications in diagnosing and treating Alzheimer’s disease (AD) [47]. Rudolph et al. utilized mirror phage display to identify the Mosd1 (monomer-specific d-peptide 1) peptide, which has the capability to degrade Aβ1–42 and diminish its associated toxicity in ex vivo experiments [48]. In addition, recent research utilizing phage display technology has resulted in the discovery of a peptide that specifically interacts with β-secretase (BACE1), effectively preventing the formation of Aβ plaques [49]. Also, researchers have engineered peptides that specifically target the receptors for advanced glycation end products (RAGE). These peptides contribute to reducing cellular degradation associated with Aβ by decreasing BACE1 activity and suppressing apoptosis [50]. Employing similar approaches, anticalins derived from human lipocalin have proven effective in neutralizing and reducing the aggregation of Aβ [51].

Researchers discovered an approach to prevent the aggregation of the Aβ peptide by targeting its fibril-dependent nucleation process. Utilizing phage display, they produced scFv fragments that exhibited a strong affinity for Aβ fibrils, hence preventing Aβ aggregation by impeding secondary nucleation [52]. In addition, peptides from both healthy controls and AD patients were identified in clinical research using phage display. The ability of these peptides to identify inflammatory cytokines linked to AD has led to their use as biomarkers in the diagnosis of the disease [53].

The application of highly efficient screening methods using T7 phage libraries has facilitated the identification of autoantibodies for AD. This approach has enabled the detection of markers in both serum and cerebrospinal fluid, offering a new avenue for diagnosis [54]. Utilizing T7 phage libraries to isolate particular peptides including anthrax toxin receptor 1, nuclear protein 1, glycogen phosphorylase, and olfactory receptor 8J1 has been suggested in a recent investigation. These peptides have been proposed as promising markers for the diagnosis of AD [55]. Likewise, in mouse models of AD, the cyclic peptide DAG, developed by phage display, has been shown to bind to various peptides and highlight the overexpression of pathogenic molecules, including the connective tissue growth factor [56]. Additionally, the techniques of phage display and hybridoma have been employed to identify tau antibodies in patients with AD [57]. Studies highlight the importance of choosing target states (monomeric or aggregated Aβ) in screening to yield peptides with diverse effects on Aβ aggregation [58]. Zhang et al. successfully synthesized an Aβ oligomer binding peptide designated KM (consisting of the amino acid sequence K-S-I-L-R-T-S-I-R-H-T-H) in addition to a brain-targeting peptide with the amino acid sequence I-T-P-T-R-K-S (referred to as IS). Following this, the aforementioned peptides were incorporated in order to produce a bifunctional nanoparticle, known as IS@NP/KH, which was specifically engineered to transport the Aβ1–42 oligomer binding peptide to the brain. This nanoparticle showed therapeutic potential by improving cognitive performance and reducing Aβ plaques in AD model mice (Figure 3) [59]. Numerous target antigens have been produced using mirror-image phage display, an efficient and straightforward method of isolating peptide ligands (Figure 4A) [60]. A recent study utilized a randomized 12-mer peptide Ph.D.™ library and mirror-image phage display to isolate D-peptides that specifically targeted the whole length of the Tau protein. The D-peptides generated exhibited an outstanding ability to inhibit the aggregation of the Tau protein [61]. Another peptide, D3, obtained through mirror image phage display using D-amino acids, exhibited the ability to inhibit Aβ aggregation, improve cognitive performance, and reduce Aβ plaques in AD model mice. The study emphasizes the potential of mirror image phage display in creating D-peptides for AD treatment, highlighting their stability and ability to penetrate the blood–brain barrier (Figure 4B–F) [62]. Regarding future directions, there are plans to explore various phage-displayed peptide libraries and implement modifications to delve deeper into understanding the inhibitory effects on Aβ aggregation and AD. 

### 3.2. Inflammatory and Autoimmune Diseases

#### 3.2.1. Rheumatoid Arthritis

Rheumatoid arthritis (RA), an autoimmune disorder primarily affecting peripheral synovial joints, has prompted the exploration of immunogenic profiles using various high-throughput techniques [34,35,36]. Recent advancements in phage display technology have identified potential biomarkers for seronegative RA, with a high-throughput cDNA phage display library screening from RA patients’ synovial tissue revealing novel autoantibody biomarkers. This methodology, applied in the CareRA trial, holds promise for discovering theranostic markers to enhance RA treatment [33,37].

In a separate investigation using programmable phage display technology, specifically Phage ImmunoPrecipitation Sequencing (PhIP-Seq), researchers characterized the reactivities of anti-citrullinated protein antibodies (ACPA) associated with RA [63]. Hypothesized to play a role in RA pathogenesis by binding to citrullinated proteins in synovial joints [64], The study identified a phage-fused peptide labeled as M12, which demonstrated high immunoreactivity against RA sera. M12, synthesized from the selected clone, exhibited significant diagnostic potential, distinguishing RA patients from those with systemic lupus erythematosus (SLE), ankylosing spondylitis (AS), and healthy controls (HCs) with notable specificity and sensitivity [65].

Examining the therapeutic potential of the mimotope PBP (PGE2 receptor EP4) for adjuvant-induced arthritis (AA), identified through phage display techniques, the study suggests PBP as a promising therapy for RA. The findings indicate its potential to mitigate joint inflammation and destruction [66]. Additionally, a study employing phage-display methods cloned high-affinity human monoclonal anti-glucose-6-phosphate isomerase (GPI) IgGs from an RA patient, revealing somatic mutations indicative of an antigen-driven, affinity-matured response. Immunohistochemistry of RA synovium unveiled elevated GPI concentrations, suggesting a potential mechanism for antibody-induced joint disease [67].

Furthermore, a study using a random peptide combinatorial phage library and phage display investigated the etiology of RA, showcasing diverse antibody reactivity within a family with rheumatic manifestations. Independent of polyreactive IgM antibodies or rheumatoid factors, this research highlights the utility of phage display in probing antibody responses and unraveling connections between autoimmune diseases [68]. 

#### 3.2.2. Multiple Sclerosis

Characterized by inflammation and autoimmune processes within the nervous system, multiple sclerosis (MS) ranks as the second most common neurological disability affecting individuals in their youth and middle age. In this investigation, phage display technology was employed to identify potential autoantigens linked to multiple sclerosis, revealing a complex of 14 antigens. Notably, DDX24 and TCERG1, implicated in RNA modification and regulation, emerged as promising candidates, indicating their potential as markers for specific disease activity states in multiple sclerosis [42]. The study utilized the Serological Antigen Selection (SAS) cDNA phage display method to identify autoantibody-inducing immunogenic targets in the serum of multiple sclerosis (MS) patients. The optimization process involved alternating positive selection with MS serum and negative selection with healthy control serum, effectively preventing the overgrowth of IgG-displaying phage clones and facilitating the identification of potential MS-related antigens [69].

In the investigation using an experimental autoimmune encephalomyelitis (EAE) rat model mimicking multiple sclerosis, the phage display strategy pinpointed the circular peptide CLSTASNSC as a proficient binder to inflammatory regions in the central nervous system. This peptide, demonstrating effective binding to activated human brain endothelial cells under inflammatory conditions, stands as a potential candidate for further exploration into molecular alterations in inflammatory lesions associated with multiple sclerosis [41]. The study addressed the challenge of investigating oligoclonal antibodies in the cerebrospinal fluid (CSF) of multiple sclerosis (MS) patients by employing a phage-displayed random peptide library. Identified peptides, mimicking various natural epitopes, were recognized by CSF antibodies in MS patients, suggesting individual-specific immunodysregulation rather than a stereotyped response to a single antigen/agent in the CSF antibody repertoire [70].

Recombinant antibodies (rAbs) were generated from over-represented IgG sequences in multiple sclerosis (MS) cerebrospinal fluid (CSF), and phage display revealed specific peptide sequences. Inhibition assays confirmed the peptides’ specific binding to the antibody’s antigen-binding site, indicating that epitopes/mimotopes identified by MS rAb could offer insights into disease-relevant antigens [71]. The study employed phage immunoprecipitation sequencing to screen 298 antibody repertoires, identifying individual-specific autoantibody fingerprints, including those associated with diseases like type 1 diabetes (T1D), multiple sclerosis, and rheumatoid arthritis. The data highlights the prematurely polyautoreactive phenotype in T1D patients and uncovers novel antibody specificities associated with multiple sclerosis and rheumatoid arthritis, contributing to a better understanding of autoimmunoreactivities in health and disease [72].

#### 3.2.3. Systemic Lupus Erythematosus

Systemic lupus erythematosus (SLE) is a persistent autoimmune condition marked by the immune system mistakenly targeting healthy cells and tissues, resulting in inflammation and diverse symptoms throughout the body [39]. Sulfatide, present in various tissues, induces the production of antibodies in individuals with demyelinating peripheral neuropathy, HIV infection, and lupus. This has potential implications for clinical symptoms. Utilizing a phage-display library from lupus patients, researchers isolated sulfatide-reactive antibodies, revealing specific binding patterns and suggesting a role in the symptoms and pathophysiological processes of lupus [40].

In the quest for potential biomarkers for systemic lupus erythematosus (SLE), researchers identified and validated four highly diagnostic peptides (SLE2018Val001, SLE2018Val002, SLE2018Val006, and SLE2018Val008). This study, involving 306 participants across healthy, SLE, and other autoimmune-related disease groups, employed a statistical analysis protocol and enzyme-linked immunosorbent assays (ELISA) for validation. The peptides were discovered through a phage-displayed random peptide library and deep sequencing [38]. 

Sun et al. made a significant discovery of a peptide pattern in systemic lupus erythematosus (SLE) that mimics the antigenic and immunogenic epitope of double-stranded DNA. This identified pattern is noteworthy because anti-dsDNA and anti-ssDNA antibodies, upon binding to glomerular antigens, contribute to immune deposits, playing a crucial role in kidney damage among SLE patients [73].

### 3.3. Chronic Non-Communicable Diseases

Chronic non-communicable diseases (NCDs) are responsible for the majority of worldwide fatalities, particularly prevalent in low- and middle-income nations. Key contributors to these diseases include inadequate dietary habits, sedentary lifestyles, smoking, and excessive alcohol consumption. The prevalence of these risk factors is on the rise, primarily due to the processes of urbanization and globalization.

#### 3.3.1. Diabetes Mellitus

Elevated blood glucose levels are the hallmark of diabetes mellitus (DM), a metabolic condition that can develop due to insufficiency in insulin synthesis or its action, or both [74]. It is primarily classified into two types, with both exhibiting similar systemic effects, including nephropathy and cardiomyopathy [75]. In a recent study, a therapeutic strategy leveraging phage display technology was developed with the aim of reducing the risk of thrombosis specific to diabetes [76]. Phage display has also been used to investigate and examine diabetes-related molecules, including the insulin-regulated glucose transporter type 4 [77]. 

The specific targeting of alpha-amylase in diabetic patients has been suggested as a result of its function in the hydrolysis of dietary starch into simple sugars. Accordingly, numerous methodologies have been devised to synthesize alpha-amylase inhibitors for therapeutic purposes for people with diabetes. Ngoh and colleagues utilized phage display technology to identify new anti-amylase agents from pinto beans in their research. Their study identified a clone, SyP9, exhibiting the most potent inhibitory activity against alpha-amylase. Specifically, the peptide showed a strong affinity towards specific residues in binding sites of alpha-amylase [78].

T regulatory cells and chimeric antigen receptors (CARs) have been specifically directed toward certain antigens in order to achieve therapeutic benefits. Using phage display methods, Tenspolde et al. recently documented the development of scFvs and insulin-specific CARs. Insulin-specific bacteriophages were isolated from ScFv HAL9 and HAL10 phage libraries. Using a retroviral CAR vector, these phages were subsequently employed to create insulin-specific chimeric CARs. As shown in Figure 5A, this vector included co-stimulation and T-cell activation domains like CD28 and Foxp3. The generated CAR-Treg cells altered the specificity of T cells, enhancing their affinity for insulin (Figure 5B) [79]. 

Despite exenatide being a common treatment for type II diabetes, its effectiveness is limited due to its short half-life. Research employing phage display technology successfully isolated a peptide that binds strongly to human serum albumin, enhancing the half-life of exenatide by four times. Additionally, this interaction was associated with an improvement in glucose tolerance [80]. Phage display libraries were employed by Demartis et al. to isolate peptides that function as agonists for glucagon and glucagon-like receptors. On the basis of the substantial involvement of these receptors in diabetes and obesity, they hypothesized that these peptides might be advantageous in the treatment of these disorders [81]. 

Recently, autoantibodies against ZnT8 (zinc transporter protein 8), which is released by pancreatic beta islets, have been studied as a potential indicator of type 1 diabetes [82]. The scFv phage display library was used to isolate ZnT8-specific antibodies that exhibited exceptional specificity for beta cells, making them suitable for immunodiagnostic purposes [83].

The neuropeptide pituitary adenylate cyclase-activating polypeptide (PACAP) has shown promise in the treatment of diabetes [84]. The peptide derived from PACAP, which comprises dipeptidyl peptidase IV cleavage peptide, factor Xa, and a 7-mer albumin binding peptide, increases insulin secretion and decreases blood sugar via preferential binding to vasoactive intestinal peptide receptor 2 (VPAC2) [85]. In some investigations, urinary and plasma L-carnitine levels are reduced in patients with type 2 diabetes [86]. A novel antibody fragment comprised of L-carnitine-like binding properties was identified in the Tomlinson I and J scFv libraries and could potentially utilized as a diagnostic tool for determining L-carnitine levels [87].

#### 3.3.2. Gastrointestinal Disorders 

Phage display technology is increasingly being used in several gastrointestinal tract organs, including the liver, stomach, and colon, for diagnostic and therapeutic applications. Antibodies that target *Helicobacter pylori (H. pylori)* vacuolating cytotoxin A (VacA) were recently isolated using the scFv phage library. The diagnostic potential of these antibodies was demonstrated by the high affinity of their complementarity determining regions (CDR) toward VacA toxin [88]. In addition, a similar investigation has developed ArsS antagonistic peptides that exhibit the ability to obstruct acid-sensing signal transduction, thereby presenting an unfavorable condition for *H. pylori* proliferation [89]. Facchin and colleagues obtained peptides from patients diagnosed with ulcerative colitis and Crohn’s. These peptides were subsequently incorporated into nano assemblies that mimic viruses, with each nanoparticle comprising 400 peptides. Subsequently, they employed colorimetric assays and immunofluorescent staining techniques to differentiate between samples from Crohn’s disease and ulcerative colitis [90]. Furthermore, a 7-mer phage library was exploited by Cardona-Correa and colleagues to uncover peptides that are implicated in the pathogenic process of the lethal Bacillus anthracis toxin [91]. Infections with the hepatitis B and C viruses are two of the most common causes of chronic liver disease globally [92]. A screening scFv phage display library screening system was developed by Yokokawa and colleagues for the identification of novel HCV-specific antibodies against the E2 envelope glycoprotein. It was discovered that the created scFv exhibited pan-genotypic neutralizing ability against HCV infection and was capable of targeting conformational epitopes of the E2 glycoprotein [93]. Several studies have revealed that pre-S1 and pre-S2, which are found on the outside of the HBV envelope, play a crucial role in HBV infection. With the utilization of phage display technology, a functional anti-pre-S2 scFv and a human anti-pre-S1 mAb have been successfully engineered for the purpose of preventing and treating HBV infection [94,95]. Another experiment employed a phage display to create chimeric Fab against HBeAg. A chimeric Fab against HBeAg demonstrated enhanced specificity and sensitivity than existing commercially available kits in the context of ELISA [96].

#### 3.3.3. Cardiovascular Diseases 

Phage display technology has garnered significant research interest due to its potential to detect cardiovascular disease at an early stage. A 12-mer peptide specifically targeting rat troponin I has been discovered, capable of recognizing the early phases of acute myocardial infarction [97]. By employing the scFv library ETH-2-Gold, antibodies targeting junction plakoglobin and its isoforms in atherosclerotic plaques were also discovered [98]. A recent study by Hemadou et al. examined the feasibility of employing a scFv library for in vivo phage display in order to isolate antibodies that are specific to atherosclerosis. Phages were administered into the atherosclerotic lesions of rabbits with the intention of conducting in vivo biopanning. The scFv-phages demonstrated an exceptional level of specificity towards atherosclerosis biomarkers, as determined by flow cytometry analysis [99]. The identification of galectin 3 in human and preclinical atherosclerosis plaques has also been suggested in a recent investigation [100].

Additionally, studies have been conducted on Kawasaki disease, which is associated with vascular inflammation and is associated with a variety of cardiovascular difficulties in children under the age of five. In a recent clinical investigation, serum from patients diagnosed with Kawasaki disease having viral causes and those with unknown causes were analyzed using phage immunoprecipitation sequencing. Nevertheless, this research lacked unique data due to the intricate nature of the illness [101]. Moreover, anti-HDL (high-density lipoprotein) antibodies have been produced through the biopanning technique using HDL derived from patients diagnosed with coronary artery disease. The ability of these antibodies to bind to apolipoproteins A-1 and 2 demonstrates the diagnostic utility of the phage display technology [102].

#### 3.3.4. Renal Disease

Using a nanobody library based on phage display, an antibody was produced targeting Cystatin C, a significant indicator for estimating glomerular filtration rate [103]. Subsequently, the anti-cystatin C nanobody was incorporated into an immunosensor made of titanium dioxide nanotubes, which showed exceptional sensitivity in detecting cystatin C in human serum samples [104]. Kidney Injury Molecule-1 (KIM-1), which is elevated in injured tubular epithelial cells, is a different antigen that has been targeted using phage display biopanning in renal disease. The peptides identified through this process have displayed a high binding affinity for the KIM-1 antigen, underscoring their potential applicability in diagnosing and treating both tumors and kidney injuries [105]. Kim et al. recently conducted a study where they effectively used the scFv phage library to discover an agonist antibody that targets cMet, a receptor for hepatocyte growth factor. Having high sensitivity for cMet, the antibody effectively improved cell migration and proliferation, leading to a reduction in kidney fibrosis. The observed effects were characterized by an elevation in the levels of Bcl-2 and E-cadherin, along with the downregulation of Bax-2, fibronectin, and collagen 1 [106].

#### 3.3.5. Chronic Obstructive Pulmonary Disease

Chronic obstructive pulmonary disease (COPD) is a progressive respiratory condition characterized by persistent airflow limitation, often associated with chronic bronchitis and emphysema. The research findings reveal a modified glycosaminoglycan composition in the diaphragms of individuals with COPD, potentially impacting cellular physiology and interactions with growth factors. This modification may be associated with a reduction in levels of contractile proteins [107]. Notably, various peptide drugs obtained through phage display, such as Fuzeon for HIV treatment, Goserelin for breast and prostate cancer, and DX-890 for pulmonary conditions like cystic fibrosis and COPD, are either widely employed in clinical settings or currently undergoing clinical trials [108].

In a distinct investigation, the study explores the diagnostic significance and regulatory mechanisms of genes related to macrophage polarization in COPD using bioinformatics analysis, considering the potential influence of phages. The results highlight GEM as a potential biomarker linked to COPD and macrophage polarization, indicating a possible connection between phage-related factors and the diagnosis of COPD [109].

#### 3.3.6. Chronic Kidney Disease

Chronic kidney disease (CKD) is a long-term condition characterized by the gradual loss of kidney function over time. A novel method using phage-display peptide libraries identified distinct peptide motifs (ELRGD(R/M)AX(W/L)) for basolateral surfaces of rat proximal convoluted tubules (PCT) and cortical collecting ducts (CCD), indicating varied membrane receptor expression. The recovered linear peptide demonstrated 16-fold selectivity for phage binding to CCD over PCT, emphasizing the importance of native cell systems in identifying renal epithelial cell surface ligands, with potential applications in drug delivery or gene transfer for kidney-specific targeting [110].

### 3.4. Other Diagnostic and Therapeutic Application

Phage display technology can be used to select anti-venom antibodies as well as antibodies against toxins produced by microorganisms and plants. One investigation has discovered that peptides against Phospholipase A_2_ (PLA_2_) inhibit PLA_2_ activity in the venom of western cottonmouth [111]. Another study used phage display technology to produce IgG monoclonal antibodies in order to neutralize the dendrotoxin-mediated neurotoxicity of venom derived from a black mamba snake [112]. In terms of plant toxins, Pelat et al. exploited phage display to design an anti-ricin scFv that could inactivate ricin biological activity and, therefore, could be used as a first-line therapy for ricin poisoning [113]. The application of the phage display approach to create a vaccine against systemic candidiasis was recently proposed in a mouse model [114]. Several experiments have employed phage display to generate antibodies against bacterial toxins such as scFv antibodies against botulinum and alpha-toxin or VHH antibodies against anthrax toxin and Clostridioides Difficile toxin [115]. In a very recent study, Xu et al. used phage display to produce a scFv-Fc antibody that neutralized the α-hemolysin toxin of Staphylococcus aureus. The obtained scFv-Fc demonstrated considerable binding affinities to α-hemolysin, suggesting that it could be a viable candidate to prevent MRSA (Methicillin-resistant Staphylococcus aureus) infection [116]. Moreover, in one research, phage display was utilized to create 68Ga-labeled peptides that could be used in positron emission tomography (PET) scans to identify *Staphylococcus aureus* biofilms [117]. Table 1 summarizes a list of approved antibodies developed by the phage display technique.

## 4. Conclusions

Phage display technology has become significantly prevalent across various fields, notably in the identification of biomarkers, the study of molecular interactions, and the enhancement of drug delivery mechanisms. Furthermore, this technology is predicted to be a potent framework for leading breakthrough therapy and diagnostics in the long term. Phages have the ability to overcome environmental stresses that are present in experimental settings including ultraviolet (UV) radiation [145,146], adverse temperature and pH conditions [147], and proteolytic enzymes [148], making them promising candidates for biosensing under these severe conditions [149]. Due to its resilience and remarkable stability, it is considered the benchmark for monoclonal antibody isolation. A total of 80 monoclonal antibodies have been extracted by phage display technology and are currently undergoing clinical trials, with 7 approved products, so far.

Despite the numerous potential benefits of antibody phage display technology, certain intrinsic restrictions still exist. For example, several characteristics influence the size of the libraries, which is important for reaching the intended affinities. This can be a pricey and tedious operation. Additionally, phages can only produce fragments of antibodies (like scFv and Fab), therefore it could be challenging to create conjugated antibodies owing to their small size. The benefits of phage display technology outweigh the drawbacks.

## 5. Future Perspective

Due to the extensive applications of phages, ongoing research endeavors may be dedicated to integrating this technology with other fields such as nanotechnology, therapeutic pharmaceuticals, diagnostic tools, and machine learning to optimize its advantages. Engineered phage particles combined with nanoparticles may offer a more secure and efficient means of delivering therapeutic cargo to specific cells associated with certain health conditions. Additionally, artificial intelligence (AI) holds the potential to transform phage display libraries. Utilizing deep sequencing data [150] and machine learning [151] can help identify sequence spaces containing functional variants enriched in deep sequencing data. Consequently, this approach aids in recognizing, categorizing, and quantifying features relevant to diagnostics and therapy [150,151,152].

By leveraging phage libraries to screen for peptides or antibodies targeting specific disease-related molecules, we can uncover novel diagnostics and therapeutics. For infectious diseases, this approach offers avenues for discovering new antibiotics and antiviral agents, aiding in rapid detection and treatment. Similarly, in autoimmune disorders, phage display enables the identification of mimotopes that mimic self-antigens, paving the way for innovative immunotherapies. Moreover, in neurological disorders, cardiovascular diseases, metabolic disorders, and rare diseases, phage display presents opportunities for developing targeted therapies and diagnostic tools tailored to the unique characteristics of each condition. Looking ahead, the future of phage display in non-cancerous diseases is poised for transformative advancements. With ongoing progress in high-throughput screening techniques and computational modeling, we can expedite the discovery and optimization of phage-derived molecules for clinical applications. Furthermore, the integration of phage display with other cutting-edge technologies, such as CRISPR-based genome editing and single-cell sequencing, holds the potential to unlock new insights into disease mechanisms and therapeutic targets. Envisioning the future of diagnostic applications, user-friendly, secure, and cost-effective phage-based biosensors could evolve into lab-on-a-chip devices, becoming readily available in households for the early detection of various diseases expressing specific biomarkers.

## Figures and Tables

**Figure 1 molecules-29-03002-f001:**
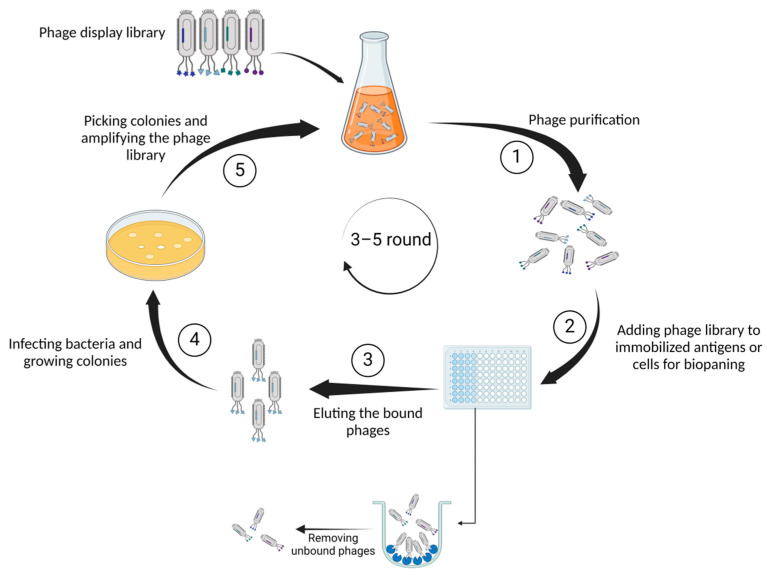
Illustrates the fundamental steps of phage display technology. First, (1) genetically identical phage populations are isolated from an environmental sample, ensuring a pure sample for detailed characterization through serial dilution. Next, (2) diverse phages presenting unique peptides or proteins on their surfaces are exposed to antigens immobilized on a surface. Capturing the phages binding to antigens facilitates the selection and isolation of those with specific affinity. Subsequently, (3) the bound phages are detached from the antigens for collection and further analysis. In the following phase (4), phages employ bacterial machinery to replicate, generating numerous copies or colonies. Finally, (5) specific bacterial colonies containing phages of interest are isolated and grown to augment the overall quantity of the phage library.

**Figure 2 molecules-29-03002-f002:**
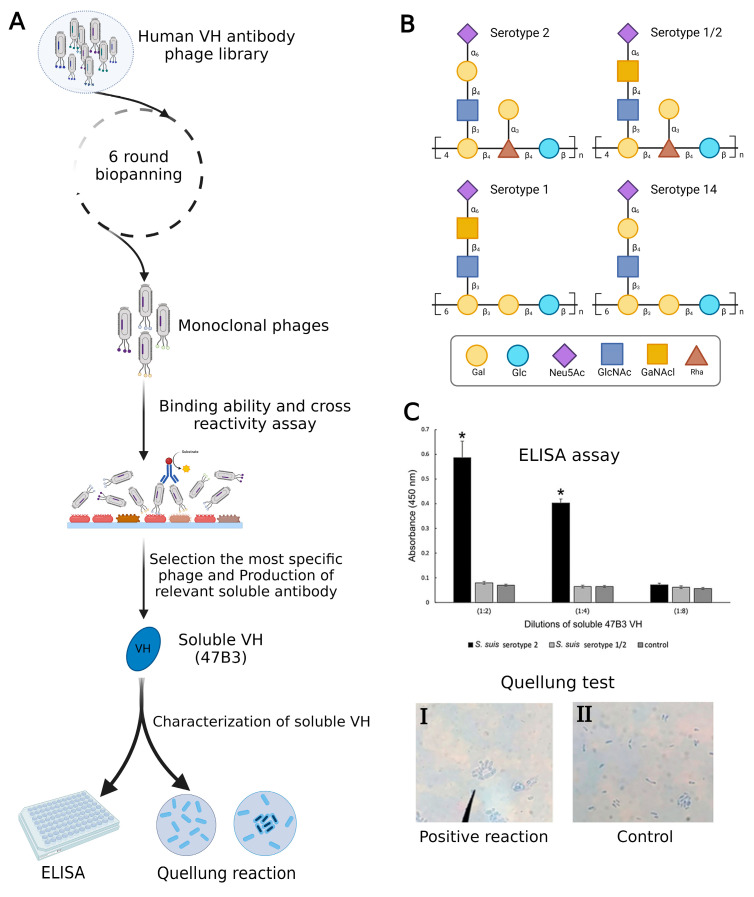
(**A**) Summary of production VH antibody against CPS of *Streptococcus suis* serotype 2 and assay methods. (**B**) Comparison of CPS repeating units among serotypes of *S. suis.* (**C**) Represented cross-reactivity and binding ability of pure 47B3 VH against *S. suis* serotypes 2 and 1/2. By using ELISA, soluble 47B3 VH demonstrated selective binding toward serotype 2 and no cross-reactivity with serotype 1/2. Swelling of the capsule when bound to soluble 47B3 VH represented a positive quellung reaction (**I**) and as a control, bacterial cells without treatment with 47B3 VH remained with intact capsule (**II**) [30]. * *p* < 0.05 indicates a statistically significant difference. Abbreviations: D-galactose (Gal), D-glucose (Glc), N-acetyl-d-neuraminic acid (Neu5Ac), L-rhamnose (Rha), N-acetyl-d-glucosamine (GlcNAc), and N-acetyl-d-galactosamine (GalNAc).

**Figure 3 molecules-29-03002-f003:**
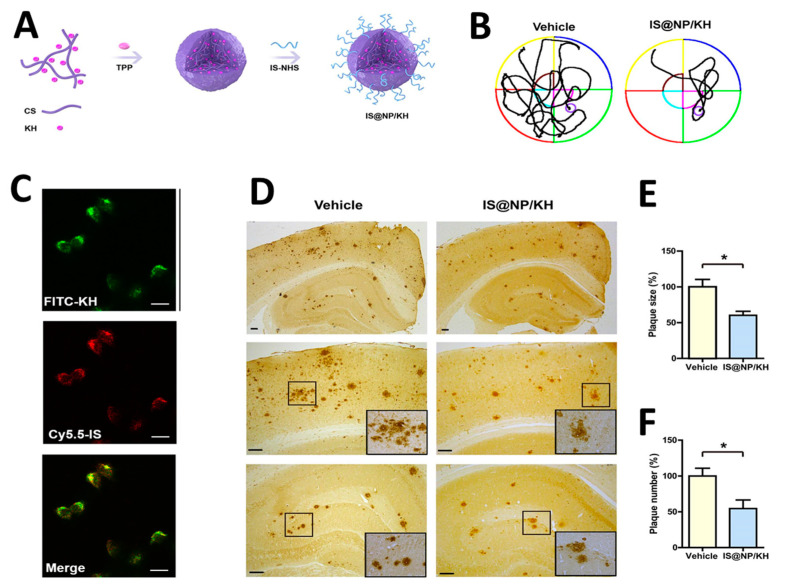
(**A**,**B**) A schematic displaying the processes involved in preparing IS@NP/KH. (**C**) Confocal microscope images demonstrate the effective uptake of IS@NP/KH by immobilized N2a-sw cells. (**D**) Treatment with IS@NP/KH reduced amyloid plaque deposition in the brains of APP/PS1 mice. Shown here are representative images of Aβ plaques. (**E**,**F**) Quantification revealed a reduction in both the number and size of Aβ plaques in the brains of APP/PS1 mice following treatment with IS@NP/KH. Reproduced with permission from, Copyright 2021, American Chemical Society [59]. * *p* < 0.05 indicates a statistically significant difference.

**Figure 4 molecules-29-03002-f004:**
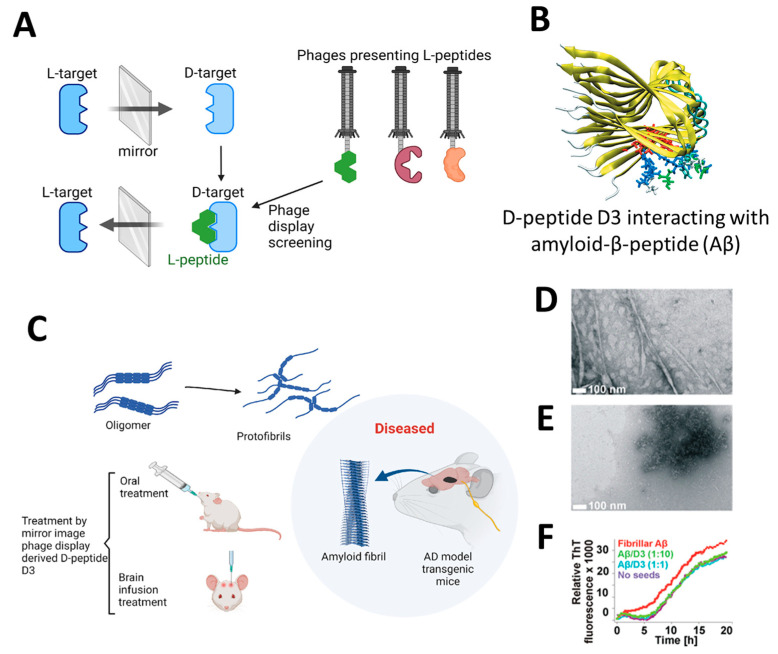
(**A**) Illustrates the process of mirror-image phage display using D-amino acid proteins as targets [61]. (**B**) Depicting the molecular structure of the Aβ targeting D-enantiomeric amino acid peptide targeting Aβ. (**C**) Derived from mirror-image phage display, the D-enantiomeric peptide D3 can enhance the cognitive condition of transgenic AD mice and decrease the accumulation of Aβ plaques. (**D**,**E**) The electron microscopy images shown here depict Aβ samples using the D-enantiomeric peptide D3, with D3 absent (**D**) and present (**E**). (**F**) Displays the Aβ fibril observed with ThT fluorescence detection. Aβ fibrils and Aβ-D3 co-aggregates were generated using specific conditions. The time-dependent ThT fluorescence was measured in various quantities of D3. Reproduced with permission from, Copyright 2010, American Chemical Society [62].

**Figure 5 molecules-29-03002-f005:**
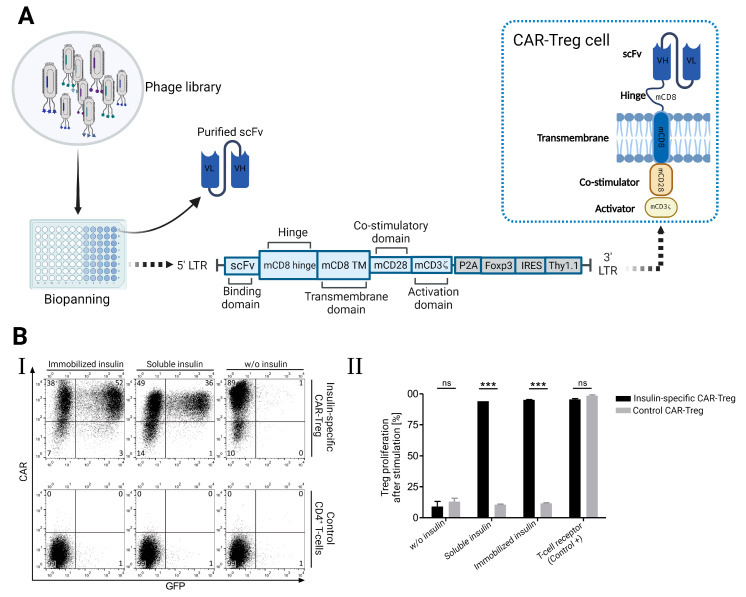
(**A**) Illustrates the application of the phage display technique in the generation of T-regulatory cells that express insulin-specific chimeric receptors (CAR). (**B**) CAR-Treg cells showed high specificity toward insulin according to the upregulation of GFP expression (**I**) and strong proliferation in the presence of insulin (**II**) [79]. *** *p* < 0.001 indicates a statistically significant difference, ns: not significant.

**Table 1 molecules-29-03002-t001:** A list of antibodies that have been approved employing phage display technology, respecting the approval date.

Antibody Name	Format	Target	Indications (Year of Approve)	Reference
Tralokinumab	IgG4-λ	IL13	Atopic dermatitis (2021)	[118]
Ramucirumab	IgG1-κ	VEGFR2	Hepatocellular carcinoma (2019) Colorectal cancer (2015) Gastric cancer, non-small cell lung cancer (2014)	[119,120,121]
Atezolizumab	IgG1-κ	PD-L1	Breast cancer (2019) Urothelial bladder cancer (2017) Non-small cell lung cancer (2016) Urothelial carcinoma (2016)	[122,123,124,125]
Avelumab	IgG1-λ	PD-L1	Renal cell carcinoma (2019) Merkel-cell carcinoma, metastatic urothelial carcinoma (2017)	[126,127]
Ixekizumab	IgG1-κ	IL17A	Ankylosing spondylitis (2019) Psoriatic arthritis (2017) Psoriasis (2016)	[128,129,130]
Caplacizumab	VHH	VWF A1 domain	Acquired thrombotic thrombocytopenic purpura (2018)	[131]
Moxetumomab	Fv-PE38	CD22	Hairy cell leukemia (2018)	[132]
Emapalumab	IgG1-λ	Interferon-gamma	Hemophagocytic lymphohistiocytosis (2018)	[133]
Guselkumab	IgG1-λ	IL23	Psoriasis (2017)	[134]
Lanadelumab	IgG1-κ	Plasma kallikrein	Hereditary angioedema (2017)	[135]
Ranibizumab	Fab-IgG1-κ	VEGFA	Diabetic retinopathy (2017) Visual impairment due to choroidal neovascularization (2016) Diabetic macular edema (2012) Macular edema following retinal vein occlusion (2010) Neovascular age-related macular degeneration (2006)	[136,137,138,139,140]
Necitumumab	IgG1-κ	EGFR	Non-small cell lung cancer (2015)	[141]
Raxibacumab	IgG1-λ	Anthrax PA, Bacillus anthracis	Inhalation anthrax (2012)	[142]
Belimumab	IgG1-λ	BLyS	Systemic lupus erythematosus (2011)	[143]
Adalimumab	IgG1-κ	TNFAα	Rheumatoid arthritis (2002)	[144]

Abbreviations. IL: interleukin, VEGFR: vascular endothelial growth factor receptor, VWF: von Willebrand factor, VEGFA: vascular endothelial growth factor A, EGFR: epidermal Growth factor receptor, PA: protective antigen, BLyS: B-lymphocyte stimulator, TNFAα: tumor necrosis factor-alpha.

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
