# Peer review of "Phage Display Technology in Biomarker Identification with Emphasis on Non-Cancerous Diseases"

_molecules, 2024, doi:10.3390/molecules29133002_

Round 1

Reviewer 1 Report

Comments and Suggestions for Authors

This review introduces the recent progress of phage display technology, emphasizing the importance of phage display technology in medical diagnostic and therapeutic fields. The content of this review is interesting and the manuscript is well written.

Some minor revisions are recommended as follows:

Abstract is too simple and the key findings of this review are not introduced.

Introduction: This section should introduce the general information on phage display technology, and the existing reviews about phage display technology. The authors should point out the importance of this review as well as the differences of this review with existing reviews.

“1.1. Natural and synthetic peptide library, 1.2. Antibodies-based libraries, 1.3. Biopanning and 1.4. The diagnostic effectiveness of phage display technology” This contents should not be included in introduction but should be introduced in a separate section, giving a title as “2. The fundamental steps of phage display technology”.

“2. Non-cancerous diseases” The title is too simple and cannot reflect the content of this section.

When introducing each non-cancerous disease, I suggest the authoring adding a figure to illustrate the application of phage display technology in diagnose and therapy of each disease, because the pathogenesis is totally different.

Future perspectives are too simple and the author should give more guidelines for future studies.

Comments on the Quality of English Language

Minor editing of English language required

Author Response

Dear Reviewer 1, 

We sincerely appreciate your thorough review of our manuscript on the recent progress of phage display technology. Your constructive feedback has been invaluable in refining our work to ensure its clarity and effectiveness.

In response to your suggestions, we have made the following revisions, all of which are now marked in blue within the updated manuscript:

  • Abstract Enhancement: We have expanded the abstract to include key findings from our review, providing a more comprehensive overview of the content. 

  • Introduction Clarification: The introduction now provides a broader context of phage display technology, highlighting its significance in medical diagnostics and therapeutics. Additionally, we have outlined the unique contributions of our review compared to existing literature.

  • Reorganization of Content: The fundamental steps of phage display technology have been moved to a separate section titled "The Fundamental Steps of Phage Display Technology" for better clarity and organization. Following is the new outline in the manuscript:

  1. Introduction
  2. The fundamental steps of phage display technology

2.1. Natural and synthetic peptide library

2.2. Antibodies-based libraries 

2.3. Biopanning

2.4. The diagnostic effectiveness of phage display technology 

  1. Applications of phage display in non-cancer diseases

3.1. Degenerative diseases

3.1.1. Alzheimer’s disease

3.2. Inflammatory and Autoimmune Diseases

3.2.1. Rheumatoid arthritis (RA)

3.2.2. Multiple Sclerosis

3.2.3. Systemic Lupus Erythematosus 

3.3. Chronic non-communicable diseases 

3.3.1. Diabetes Mellitus

3.3.2. Gastrointestinal disorders 

3.3.3. Cardiovascular diseases

3.3.4. Renal diseases

3.3.5. Chronic Obstructive Pulmonary Disease

3.3.6. Chronic Kidney Disease

3.4 Other diagnostic and therapeutic application  

  1. Conclusion
  2. Future perspective

  • Section Title Improvement: The section titled "Non-cancerous Diseases" has been revised to “Applications of phage display in non-cancer diseases”

  1. Inclusion of Figures: We have incorporated figures (Figure-4) to illustrate the application of phage display technology in the diagnosis and therapy of various non-cancerous diseases, enhancing the clarity and understanding of the text. In addition, Figure 3 has been redrawn and updated with more description in the main text as well as the figure caption.

  1. Expansion of Future Perspectives: The section on future perspectives has been expanded to provide more comprehensive guidelines for future studies, ensuring a thorough exploration of potential research directions.

Once again, we extend our gratitude for your insightful feedback. We believe that these revisions have significantly strengthened the manuscript, and we are confident that it now provides a more robust and informative overview of phage display technology in medical diagnostics and therapeutics.

Thank you for your time and consideration.

Sincerely,

Abasalt Bahrami

On behalf of all the authors

Reviewer 2 Report

Comments and Suggestions for Authors

The review article has a substantial matching in wording with the other article ‘‘Clinical implications of phage display technique: Digging deeper in diagnostic and therapeutic grounds’’ by one of the authors. This is also evident from the iThenticate software report. This reviewer suggests first revised the manuscript thoroughly to avoid the substantial matching from the above-mentioned article before reviewing the manuscript.

Author Response

Dear Reviewer 2,

Thank you for taking the time to review our manuscript and providing positive feedback. We appreciate your insightful comments and suggestions for improving the quality of our work. Based on your feedback, we have carefully addressed each of the points raised and made revisions accordingly. 

Thank you for taking the time to review our manuscript titled "Phage Display Technology in Biomarker Identification with Emphasis on Non-Cancerous Diseases". We appreciate your thorough evaluation and constructive feedback. Your insights have been very important in improving the quality of our work.

We acknowledge the concern you raised regarding the substantial matching in wording between our review article and our preprint, "Clinical implications of phage display technique: Digging deeper in diagnostic and therapeutic grounds." Upon careful consideration of your feedback and the iThenticate software report, we have made significant revisions to address this issue. Specifically, we have meticulously reviewed the text to ensure that it is distinct from our preprint. We have rephrased and rewritten sections where similarities were identified, thereby enhancing the originality and integrity of the manuscript.

We apologize for any oversight in the initial submission and sincerely appreciate your diligence in bringing this matter to our attention.  We believe that the revisions we have made will address your feedback. 

Thank you once again for your valuable contribution to the peer review process.

Sincerely,

Abasalt Bahrami

On behalf of all the authors
